Recent updates on metabolite composition and medicinal benefits of mangosteen plant

Aizat Wan Mohd wma@ukm.edu.my
Jamil Ili Nadhirah
http://orcid.org/0000-0003-3159-6743 Ahmad-Hashim Faridda Hannim
Noor Normah Mohd
Institute of Systems Biology (INBIOSIS), Universiti Kebangsaan Malaysia (UKM) , Bangi, Selangor , Malaysia
González-Burgos Elena
Electronic publication date: 2019 Jan 31
Publication date: 2019
Volume: 7
Electronic Location ID: e6324
Received 2018 Sep 5; Accepted 2018 Dec 20
Copyright: © 2019 Aizat et al.
Copyright year: 2019
Copyright holder: Aizat et al.
License: This is an open access article distributed under the terms of the Creative Commons Attribution License, which permits unrestricted use, distribution, reproduction and adaptation in any medium and for any purpose provided that it is properly attributed. For attribution, the original author(s), title, publication source (PeerJ) and either DOI or URL of the article must be cited.
License URL: https://creativecommons.org/licenses/by/4.0/

Keywords: Manggis, Garcinia mangostana L., Natural product, Pharmaceutical, Medicine

Funding: Universiti Kebangsaan Malaysia (UKM) Research University Grant GUP-2018-122 Sciencefund grant 02-01-02-SF1237 Ministry of Science, Technology and Innovation (MOSTI) Malaysia and Fundamental Research Grant Scheme FRGS/2/2014/SG05/UKM/02/2 Ministry of Education (MOE), Malaysia This work was supported by the Universiti Kebangsaan Malaysia (UKM) Research University Grant (GUP-2018-122), a Sciencefund grant (02-01-02-SF1237) from the Ministry of Science, Technology and Innovation (MOSTI), and the Malaysia and Fundamental Research Grant Scheme (FRGS/2/2014/SG05/UKM/02/2) from the Ministry of Education (MOE), Malaysia. The funders had no role in study design, data collection and analysis, decision to publish, or preparation of the manuscript.

==============================
Background

Mangosteen (Garcinia mangostana L.) fruit has a unique sweet-sour taste and is rich in beneficial compounds such as xanthones. Mangosteen originally been used in various folk medicines to treat diarrhea, wounds, and fever. More recently, it had been used as a major component in health supplement products for weight loss and for promoting general health. This is perhaps due to its known medicinal benefits, including as anti-oxidant and anti-inflammation. Interestingly, publications related to mangosteen have surged in recent years, suggesting its popularity and usefulness in research laboratories. However, there are still no updated reviews (up to 2018) in this booming research area, particularly on its metabolite composition and medicinal benefits.

Method

In this review, we have covered recent articles within the years of 2016 to 2018 which focus on several aspects including the latest findings on the compound composition of mangosteen fruit as well as its medicinal usages.

Result

Mangosteen has been vastly used in medicinal areas including in anti-cancer, anti-microbial, and anti-diabetes treatments. Furthermore, we have also described the benefits of mangosteen extract in protecting various human organs such as liver, skin, joint, eye, neuron, bowel, and cardiovascular tissues against disorders and diseases.

Conclusion

All in all, this review describes the numerous manipulations of mangosteen extracted compounds in medicinal areas and highlights the current trend of its research. This will be important for future directed research and may allow researchers to tackle the next big challenge in mangosteen study: drug development and human applications.

Introduction

Mangosteen (Garcinia mangostana L.) belongs to the Guttiferae (syn. Clusiaceae) family, typically grown in tropical South East Asian countries such as Malaysia, Indonesia, and Thailand. Mangosteen fruit has become one of the major agricultural produce from these countries due to its high commercial value in various parts of the world including China, Japan, European, and Middle Eastern countries as well as the United States of America (www.fao.org, accessed November 2018; Table S1) (Dardak et al., 2011). The exotic appearance and unique sweet-sour taste of this fruit further enhance its appeal as a premium fruit on the shelves of most developed countries.

Mangosteen tree can reach up to six to 25 m height with lushes of leathery thick leaves canopying the tree (Fig. 1A) (Osman & Milan, 2006). Meanwhile its fruit is round with thick skin (or also called pericarp) and ripens seasonally, from green to yellow to pink spotted and finally full purple colored fruit (Fig. 1B) (Abdul-Rahman et al., 2017; Parijadi et al., 2018). The edible portion of the fruit resides within the pericarp, comprising of three to more than eight septa or also called aril, white in color and having sweet-sour taste (Osman & Milan, 2006). Its seeds also reside in one or two septa per fruit and are known to be recalcitrant, extremely sensitive to cold temperature and drying (Mazlan et al., 2018a, 2018b). The seeds of this fruit also develop apomictically without relying on sexual reproduction (Mazlan et al., 2019; Yapwattanaphun et al., 2014) as well as requiring a long period of planting before bearing (usually 7 to 9 years), which limits its agronomical improvement and cross-breeding (Osman & Milan, 2006). Furthermore, the top of the fruit is equipped with thick sepals which collectively resembles a crown, hence its popular designation, “The Queen of Tropical Fruit.” Such a designation is also commonly attributed to the plethora of medicinal benefits of this fruit as well as its unique taste (Fairchild, 1915).

Figure 1 A representative mangosteen tree grown at the experimental plot of Universiti Kebangsaan Malaysia (UKM), Malaysia (A) and a ripened mangosteen fruit (B).

Pictures are courtesy of Othman Mazlan, Institute of Systems Biology (INBIOSIS), UKM.

Mangosteen has been used in folk medicines such as in the treatment of diarrhea, wound infection, and fever (Osman & Milan, 2006; Ovalle-Magallanes, Eugenio-Pérez & Pedraza-Chaverri, 2017). Traditionally, various parts of mangosteen tree including leaves, root, and fruit are prepared by dissolving them in water or clear lime extract before usage (Osman & Milan, 2006). These days, mangosteen fruit extract is commonly commercialized as functional food or drink, with the addition of other minor components such as vitamins, which exhibits general health boost and even promoted as an anti-diabetic supplement (Udani et al., 2009; Xie et al., 2015). Furthermore, a plethora of studies have documented the fruit usages as anti-oxidant, anti-inflammatory, anti-cancer, and anti-hyperglycemic substance, perhaps due to containing bioactive compounds such as xanthones (El-Seedi et al., 2009, 2010; Ovalle-Magallanes, Eugenio-Pérez & Pedraza-Chaverri, 2017; Tousian Shandiz, Razavi & Hosseinzadeh, 2017). Interestingly, articles in this area has surged in recent years (Fig. S1) and hence, an updated review is timely to capture the current trends in mangosteen medicinal usages.

Survey methodology

Published manuscripts were obtained from various databases including Scopus, EBSCO, Web of Science, Pubmed, and Google Scholar by searching “mangosteen AND G. mangostana” in the search field. In this review, we critically cover recent articles (2016 and beyond) which is aimed to provide a comprehensive up-to-date research trend pertaining to mangosteen metabolites and their medicinal benefits.

Metabolite composition of mangosteen

Xanthone is one of the compound classes that are prevalent in mangosteen (Tousian Shandiz, Razavi & Hosseinzadeh, 2017). These metabolites have been extracted and characterized in various studies as reviewed by several publications (Ovalle-Magallanes, Eugenio-Pérez & Pedraza-Chaverri, 2017; Tousian Shandiz, Razavi & Hosseinzadeh, 2017; Zhang et al., 2017b). So far, there are more than 68 xanthones isolated from the mangosteen fruit with the majority of them are α- and γ-mangostin (Ovalle-Magallanes, Eugenio-Pérez & Pedraza-Chaverri, 2017). The molecular structure of these compounds have been elucidated (Fig. 2) and readers are directed to Ovalle-Magallanes, Eugenio-Pérez & Pedraza-Chaverri (2017) for a more descriptive review and description on these xanthones. More recently, novel xanthones have been discovered such as 1,3,6-trihydroxy-2-(3-methylbut-2-enyl)-8-(3-formyloxy-3-methylbutyl)–xanthone (Xu et al., 2016), 7-O-demethyl mangostin (Yang et al., 2017), garmoxanthone (Wang et al., 2018b) as well as mangostanaxanthone III, IV (Abdallah et al., 2017), V, VI (Mohamed et al., 2017), and VII (Ibrahim et al., 2018b) (Fig. 2). These xanthones were also implicated in various pharmaceutical properties but more studies are needed to verify their effectiveness in human applications.

Figure 2 The molecular structure of various bioactive compounds from mangosteen especially xanthones (A–X), benzophenone (isogarcinol) (Y), flavonoid (epicatechin) (Z), and procyanidin (AA).

Using High Pressure Liquid Chromatography (HPLC), Muchtaridi et al. (2017) measured the level of α-mangostin, γ-mangostin, and gartanin from different regions of Indonesia which suggest their levels can be dependent upon localities. This is interesting as xanthones may be extracted differently in different laboratories around the world, given that published manuscripts related to mangosteen and xantone extraction have originated from not just South East Asian countries, but also from United States, Japan, China, and United Kingdom (Fig. S2). Nevertheless, xanthones are known to be water insoluble and hence a few recent studies have attempted to extract such compounds by using non-polar solvents or other means possible. For instance, acetone and ethanol yielded the most amount of extracted xanthone and the highest antioxidant level compared to using other solvents such as ethyl acetate and hexane (Kusmayadi et al., 2018), whereas the extraction of α-mangostin using a single-solvent approach (methanol solvent) was more efficient compared to using an indirect solvent partitioning approach (methanol added with water then ethyl acetate extraction) as seen by the higher yield of the extracted compound (Sage et al., 2018). On the other hand, Machmudah et al. (2018) used subcritical water extraction to extract xanthones from mangosteen fruit, eliminating the need for the chemical solvents. Tan et al. (2017) and Ng et al. (2018) also showed that the aqueous micellar biphasic system they developed could also efficiently extract xanthones from mangosteen pericarp. This suggests that xanthones could be viable for human application but bioavailability studies need to be performed in the future to ascertain their delivery and efficacy. Interestingly, solubilizing α-mangostin in soybean oil (containing traces of linoleate, linolenic acid, palmitate, oleic acid, and stearate) improved the xanthone bioavailability in rats, such that the compound was found in brain, pancreas, and liver organs after 1 h treatment (Zhao et al., 2016). This signifies the potential of using oil-based formulation for increasing the bioavailability of xanthones.

While extracting and solubilizing natural xanthones have been the common strategies in mangosteen research all this while, a number of latest papers have reported the use of chemical modifications to alter the structure of xanthones. Buravlev et al. (2018) modified α-and γ-mangostin through Mannich reactions (aminomethylated at the C-4/C-5 positions) which consequently led to higher anti-oxidant activities than their original compounds. Furthermore, Karunakaran et al. (2018) showed that β-mangostin could inhibit the inflammatory response in lipopolysaccharide-induced RAW 264.7 macrophages, but this activity was not retained when the hydroxyl (OH) group at its position C-6 was replaced with acetyl or alkyl. These lines of evidence highlight the importance of certain functional groups in xanthones to confer their bioactivities including anti-oxidant and anti-inflammation.

Other than xanthones, mangosteen pericarp is also known to contain one of the highest procyanidin content, compared to other fruit such as cranberry, Fuji apple, jujube, and litchi (Zhang, Sun & Chen, 2017c). These procyanidins include monomer (47.7%), dimer (24.1%), and trimer (26%) may also contribute to the anti-oxidant capability of mangosteen extract as shown in 1,1-diphenyl-2-picrylhydrazyl (DPPH) and Ferric Reducing Antioxidant Power (FRAP) assays (Qin et al., 2017). Other phenolics such as benzoic acid derivatives (vanilic acid and protocatechuic acid), flavonoids (rutin, quercetin, cactechin, epicatechin) and anthocyanins (cyanidin 3-sophoroside) were also highly present in mangosteen pericarp (Azima, Noriham & Manshoor, 2017).

Furthermore, mangosteen compounds have also been profiled using metabolomics approach. Using GC-MS analysis, Mamat et al. (2018a) reported that mangosteen pericarp contains mainly sugars (nearly 50% of total metabolites) followed by traces of other metabolite classes such as sugar acids, alcohols, organic acids, and aromatic compounds. This study also found several phenolics such as benzoic acid, tyrosol, and protocatechuic acid which are known to possess anti-oxidative and anti-inflammatory activities (Lin et al., 2009; Ortega-García & Peragón, 2010). Another GC-MS study by Parijadi et al. (2018) reported that sugars such as glucose and fructose as well as amino acids such phenylalanine and tyrosine were significantly increased during mangosteen ripening, suggesting active metabolic process during this process. Furthermore, the study also revealed the high abundance of secondary metabolites such as 2-aminoisobutyric acid and psicose at the end of ripening process, which are possibly implicated in prolonging the fruit shelf-life (Parijadi et al., 2018). LC-MS study has also been performed in mangosteen yet the full list of metabolites has not been released (Mamat et al., 2018b).

Medicinal usages of mangosteen

In this review, medicinal benefits of mangosteen are categorized into several distinct areas including anti-cancer, anti-microbes, and anti-diabetes (Tables 1 and 2). Furthermore, its protection against damages and disorders in various human organs such as liver, skin, joint, eye, neuron, bowel, and cardiovascular tissues, either in vitro (Table 1) or in vivo (Table 2) are also evaluated and discussed.

Table 1 Summary of mangosteen medicinal usages as performed in in vitro and in silico experimentation.

Research types	Subject type	Compound name/extract used	Compound origin	Reference	
Anti-cancer	
Oral cancer	Cell lines	α-Mangostin	Commercial	Fukuda et al. (2017)	
Lung cancer	Cell lines	α-Mangostin	Commercial	Phan et al. (2018), Zhang, Yu & Shen (2017a)	
Bile duct cancer	Cell lines	α-Mangostin	Fruit pericarp	Aukkanimart et al. (2017)	
Liver cancer	Cell lines	α-Mangostin	Commercial	Wudtiwai, Pitchakarn & Banjerdpongchai (2018)	
Breast cancer	Cell lines	α-Mangostin	Commercial	Scolamiero et al. (2018)	
Anti-multidrug resistance (breast, lung, and colon cancer)	Cell lines	α-Mangostin	Commercial	Wu et al. (2017)	
Brain cancer	Cell lines	Gartanin	Fruit hull	Luo et al. (2017)	
Ovary cancer	Cell lines	Garcinone E	Fruit pericarp	Xu et al. (2017)	
Breast, lung and colon cancer	Cell lines	Garcixanthones B and C	Fruit pericarp	Ibrahim et al. (2018b)	
Breast and lung cancer	Cell lines	Mangostanaxanthone VII	Fruit pericarp	Ibrahim et al. (2018d)	
Breast and lung cancer	Cell lines	Garcixanthone A	Fruit pericarp	Ibrahim et al. (2018e)	
Breast and lung cancer	Cell lines	Mangostanaxanthone VIII	Fruit pericarp	Ibrahim et al. (2018a)	
Pancreatic cancer	Cell lines	α- and γ-Mangostin	Fruit pericarp	Kim, Chin & Lee (2017)	
Cervical cancer	Cell lines	α-Mangostin, gartanin	Fruit pericarp	Muchtaridi et al. (2018)	
Hepatocellular, breast, and colorectal cancer	Cell lines	Mangostanaxanthone IV, garcinone E, α-mangostin (all lines)	Fruit hull	Mohamed et al. (2017)	
Cervical, hepatoma, and gastric cancer	Cell lines	Garcinone E (all lines), 7-O-methylgarcinone E & α-mangostin (gastric)	Fruit pericarp	Ying et al. (2017)	
Neuroendocrine, glioma, nasopharyngeal, lung, prostate and gastric cancer	Cell lines	7-O-Demethyl mangostanin (all cancer lines), mangostanin, 8-deoxygartanin, gartanin, garcinone E, 1,3,7-trihydroxy-2,8-di-(3-methylbut-2-enyl)xanthone (neuroendocrine & glioma)	Fruit pericarp	Yang et al. (2017)	
Breast cancer	Cell lines	Ethanol extract from pericarp	Soft part of fruit peel	Agrippina, Widiyanti & Yusuf (2017)	
Lung cancer	Cell lines	Biofabrication water extracted mangosteen	Bark	Zhang & Xiao (2018)	
Anti-microbes	
Oral bacteria	Microbial culture	α-Mangostin	Fruit pericarp	Nittayananta et al. (2018)	
Dental caries prevention	Microbial culture and human tooth	α-Mangostin	Fruit rind	Sodata et al. (2017)	
Oral bacteria	Microbial culture	Ethanol: water extract	Fruit pericarp	Pribadi, Yonas & Saraswati (2017)	
Oral and gastrointestinal bacteria	Microbial culture	Methanol extract	Fruit pericarp	Nanasombat et al. (2018)	
Dental plaque	Microbial culture	Chloroform extract	Fruit pericarp	Janardhanan et al. (2017)	
Anti-bacteria and anti-biofilm	Microbial culture	α-Mangostin	Fruit peel	Phuong et al. (2017)	
Anti-bacteria and anti-biofilm	Microbial culture	α-Mangostin, ethanol extract	Fruit pericarp	Chusri et al. (2017)	
Anti-bacteria	Microbial culture and cell lines	α-Mangostin inclusion complex	Fruit hull	Phunpee et al. (2018)	
Anti-bacteria and anti-fungi	Microbial culture	α-Mangostin, 12 semi synthetic modified α-mangostin	Fruit hull	Narasimhan et al. (2017)	
Anti-bacteria	Microbial culture	Garmoxanthone	Bark	Wang et al. (2018b)	
Anti-bacterial and anti-fungal	Microbial culture	Ethyl acetate extract of leaf (lower activity in hexane and methanol extract)	Leaves	Lalitha et al. (2017)	
Anti-bacteria	Microbial culture	N-hexane:ethyl acetate	Fruit pericarp	Sugita et al. (2017)	
Anti-bacteria and anti-inflammation	Cell lines and human blood	Total extract using water and methanol	Fruit skin	Elisia et al. (2018)	
Wound healing	Microbial culture	Not described	Not described	Panawes et al. (2017)	
Anti-malaria	Microbial culture	Hexane, and ethylacetate fraction (weaker activity in water and butanol extract)	Fruit rind	Tjahjani (2017)	
Anti-dengue virus	Cell culture	α-Mangostin	Commercial	Tarasuk et al. (2017)	
Anti-diabetes	
Anti-diabetes, anti-cancer	Chicken liver	Garcinone E	Commercial	Liang et al. (2018)	
Anti-diabetes	In vitro assay	Mangostanaxanthones III and IV, β-mangostin, garcinone E, rubraxanthone, α-mangostin, garcinone C, 9-hydroxycalabaxanthone	Fruit pericarp	Abdallah et al. (2017)	
Anti-glycation	In vitro assay	Total extract using 95% ethanol	Fruit rind	Moe et al. (2018)	
Anti-diabetes	In vitro assay	Total xanthone extract using hexane	Fruit pericarp	Mishra, Kumar & Anal (2016)	
Anti-hypercholesterolemia	In silico	Epicatechin, euxanthone, and 1,3,5,6-tetrahydroxy-xanthone	Not relevant	Varghese et al. (2017)	
Liver protection	
Hepatoprotective	Cell lines	γ-Mangostin	Fruit pericarp	Wang et al. (2018a)	
Anti-oxidant	Cell lines	Isogarcinol	Bark	Liu et al. (2018)	
Skin protection	
Skin whitening	Cell lines	β-mangostin	Seedcases	Lee et al. (2017)	
Anti-oxidant (skin)	In vitro assays	Dichloromethane extract	Fruit pericarp	Chatatikun & Chiabchalard (2017)	
Photoprotective agent	Cell culture	α-Mangostin	Fruit pericarp	Im et al. (2017)	
Joint protection	
Anti-Osteoarthritis	Cell lines	α-Mangostin	Commercial	Pan et al. (2017a)	
Anti-arthritis	Cell lines	α-Mangostin	Commercial	Zuo et al. (2018)	
Eye protection	
Anti-retinal apoptosis	Cell lines	α-Mangostin	Commercial	Fang et al. (2016)	
Neuronal protection	
Enzyme inhibitor for acid sphingomyelinase, important in lung diseases, metabolic disorders, and central nervous system disease	Cell lines	α-Mangostin and modified derivatives	Fruit pericarp	Yang et al. (2018)	
Neuroprotective	Cell lines	γ-Mangostin	Fruit pericarp	Jaisin et al. (2018)	
Cardiovascular protection	
Anti-oxidant and anti-apoptosis for cardiac hypoxic injury	Cell lines	α-Mangostin	Commercial	Fang, Luo & Luo (2018)	
Anti-oxidant	Cell lines	Procyanidins	Fruit pericarp	Qin et al. (2017)	
Anti-oxidant	Cell lines	α- and γ-Mangostin and their derivatives	Dried yellow gum from fruit	Buravlev et al. (2018)	
Anti-fertility	
Pro-spermatogenic apoptosis	Cell lines and cat organs	α-Mangostin loaded into nano-carrier	Fruit pericarp	Yostawonkul et al. (2017)	
Note:

Compound origin describes the mangosteen tissue used for extraction. Compounds obtained commercially without reference to any mangosteen tissue is denoted as “commercial.” “Not described” means that the corresponding manuscript did not disclose the compound or extract used in the reported study.

Table 2 Summary of mangosteen medicinal usages as performed in in vivo experimentation.

Research types	Subject type	Compound name/extract used	Compound origin	Dosage	Ref.	
Anti-cancer	
Skin cancer	Female mice	α-Mangostin	Commercial	5 and 20 mg/kg BW	Wang et al. (2017)	
Bile duct cancer	Hamster	α-Mangostin	Fruit pericarp	100 mg/kg BW	Aukkanimart et al. (2017)	
Liver cancer	Rats	Extract powder	Fruit pericarp	200, 400, and 600 mg/kg BW	Priya, Jainu & Mohan (2018)	
Anti-microbes	
Anti-periodontitis	Human patient	Gel extract	Fruit rind	Not available	Hendiani et al. (2017)	
Anti-periodontitis	Human patient	Gel extract	Fruit pericarp	10 μL of 4% w/v	Mahendra et al. (2017)	
Dental inflammation	Guinea pigs	Not described	Fruit peel	Not available	Kresnoadi et al. (2017)	
Gingival inflammation	Rats	Not described	Fruit peel	12.5% and 25.0% w/v	Putri, Darsono & Mandalas (2017)	
Anti-diabetes	
Anti-diabetes, anti- non-alcoholic fatty liver disease (NAFLD), anti-hepatosteatosis	Male rats	α-Mangostin	Fruit pericarp	25 mg/day	Tsai et al. (2016)	
Anti-diabetes, renoprotective	Male mice	Xanthone	Commercial	100, 200, and 400 mg/kg BW	Karim, Jeenduang & Tangpong (2016)	
Anti-glycemia and anti-hepatotoxic	Male mice	Mangosteen vinegar rind (MVR) contains 69.01% alpha mangosteen, 17.85% gamma mangosteen, 4.13% gartanin, 2.95% 8-deoxygartanin, 2.84% garcinone E, and 3.22% other xanthones	Fruit rind	100 and 200 mg/kg BW	Karim, Jeenduang & Tangpong (2018)	
Anti-diabetes	Human respondents	Raw/tea	Fruit rind	Two to three times/day	Mina & Mina (2017)	
Anti-hypercholesterolemia	Male rats	Not described	Fruit rind	50, 150, 250, and 350 mg/kg BW.	As’ari & Asnani (2017)	
Liver protection	
Hepatoprotective	Male mice	α-Mangostin	Fruit pericarp	12.5 and 25.0 mg/kg BW	Fu et al. (2018)	
Hepatoprotective	Male mice	α-Mangostin	Fruit pericarp	100 and 200 mg/kg BW	Yan et al. (2018)	
Hepatoprotective	Mice	γ-Mangostin	Fruit pericarp	5 and 10 mg/kg BW	Wang et al. (2018a)	
Hepatoprotective, anti-inflammation	Male mice	Tovophyllin A	Fruit pericarp	50 and 100 mg/kg BW	Ibrahim et al. (2018c)	
Skin protection	
Anti-psoriasis (skin lesion)	Female mice	Isogarcinol	Fruit pericarp and bark	100 mg/kg BW	Chen et al. (2017)	
Photoprotective agent	Male mice	α-Mangostin	Fruit pericarp	100 mg/kg BW	Im et al. (2017)	
Joint protection	
Anti-Osteoarthritis	Male rats	α-Mangostin	Commercial	10 mg/kg BW	Pan et al. (2017a)	
Anti-inflammation, anti-arthritis	Male rats	α-Mangostin	Commercial	10 mg/kg BW	Pan et al. (2017b)	
Anti-arthritis	Male rats	α-Mangostin	Commercial	40 mg/kg BW	Zuo et al. (2018)	
Eye protection	
Anti-retinal apoptosis	Female mice	α-Mangostin	Commercial	10 and 30 mg/kg BW	Fang et al. (2016)	
Neuronal protection	
Anti-depressant	Male rats	Ethyl acetate extract	Fruit pericarp	50, 150, and 200 mg/kg	Oberholzer et al. (2018)	
Bowel protection	
Anti-colitis	Male mice	α-Mangostin	Not described	30 and 100 mg/kg BW	You et al. (2017)	
Anti-inflammatory (bowel)	Male mice	Ethanol extract	Fruit pericarp	30 and 120mg/kg BW	Chae et al. (2017)	
Cardiovascular protection	
Anti-hypertension, anti-cardiovascular remodeling	Male rats	Water extract	Fruit pericarp	200 mg/kg BW	Boonprom et al. (2017)	
Notes:

Compound origin describes the mangosteen tissue used for extraction. Compounds obtained commercially without reference to any mangosteen tissue is denoted as “commercial.” “Not described” means that the corresponding manuscript did not disclose the compound or extract used in the reported study.

BW, body weight.

Anti-cancer

α-mangostin is the largest constituent of xanthone in mangosteen pericarp extract, and hence it is well researched and applied in various cancer cell lines (Table 1). This include gastric (Ying et al., 2017), cervical (Muchtaridi et al., 2018), colorectal, hepatocellular, and breast (Mohamed et al., 2017) cancer. Furthermore, α-mangostin at a concentration of 30 μg/mL was able to reduce multicellular tumor spheroids derived from breast cancer cell lines (Scolamiero et al., 2018). The viability of human lung adenocarcinoma cell line A549 cells as well as non-small cell lung cancer cells were also negatively affected when treated with 5 μM α-mangostin (Phan et al., 2018; Zhang, Yu & Shen, 2017a). Aukkanimart et al. (2017) further demonstrated that α-mangostin-induced apoptosis in cholangiocarcinoma (bile duct cancer) cells and reduced such tumor in hamster allograft model. Human hepatocellular carcinoma (HepG2) cell lines at anoikis-resistance state (metastatic stage) was also sensitized with the treatment of α-mangostin (Wudtiwai, Pitchakarn & Banjerdpongchai, 2018). In addition, 20 mg/kg α-mangostin treatment reduced the rate of skin tumor incidence in mice (Wang et al., 2017). This suggests that α-mangostin has potent bioactivity against a diverse range of cancer cell lines and should be considered for drug developmental phase. Interestingly, α-mangostin can also inhibit ATP-binding cassette drug transporter activity, which implies that it is suitable for future cancer chemotherapy to overcome multi-drug resistance (Wu et al., 2017).

Another two bioactive xanthones from mangosteen are garcinone E and gartanin. Garcinone E has the ability to inhibit ovarian cancer cells and its action involved endoplasmic reticulum-induced stress through protective inositol-requiring kinase (IRE)-1α pathway (Xu et al., 2017). Both invasion and migration properties of the cancer cells were also significantly suppressed when treated with the compound, suggesting its potential use for anti-cancer drug (Xu et al., 2017). Furthermore, garcinone E also showed potential anti-cancer activity against cervical, hepatoma, gastric (Ying et al., 2017), breast, colorectal, and hepatocellular (Mohamed et al., 2017) cancer cell lines. Meanwhile, gartanin was demonstrated to inhibit HeLa cervical cancer cell lines (Muchtaridi et al., 2018) and suppressed primary brain tumor cells, glioma (Luo et al., 2017). The compound promoted the glioma cell cycle arrest via regulating phosphoinositide 3-kinase/protein kinase B (Akt)/mammalian Target Of Rapamycin (mTOR) signaling pathway and induced anti-migration effect via mitogen-activated protein kinases (MAPK) signaling pathway (Luo et al., 2017). Besides gartanin and garcinone E, other known mangosteen compounds such as mangostanin, 8-deoxygartanin, and 1,3,7-trihydroxy-2,8-di-(3-methylbut-2-enyl) xanthone also showed considerable anti-cancer activity against neuroendocrine and glioma cancer cell lines (Yang et al., 2017). This again suggests the applicability of isolated compounds from mangosteen for the use in anti-cancer treatment.

Recently, several newly isolated xanthones from mangosteen pericarp were shown to possess anti-cancer properties (Table 1). For example, mangostanaxanthone IV has anti-cancer activity against human breast, hepatocellular, and colorectal cell lines (Mohamed et al., 2017). Other studies showed that mangostanaxanthone VII, mangostanaxanthone VIII, garcixanthone A, B, and C were able to exert anti-proliferative activity against breast and lung cancer cell lines (Ibrahim et al., 2018a, 2018b, 2018d, 2018e). Moreover, an investigation by Yang et al. (2017) revealed that a novel isolated xanthone called 7-O-demethyl mangostanin was effective against various cancer cell lines including neuroendocrine, glioma, nasopharyngeal, lung, prostate, and gastric cancer. These lines of evidence highlight that mangosteen still has more bioactive compounds to be discovered for medicinal application.

Total extracts of mangosteen which may contain various xanthones or other metabolites have also been shown to be effective against various cancer. For instance, total pericarp extract of mangosteen was able to protect rat liver from cancer-induced diethylnitrosamine (DEN) chemical (Priya, Jainu & Mohan, 2018). Agrippina, Widiyanti & Yusuf (2017) further observed that cellulose biofilm soaked with mangosteen pericarp extract was capable of killing T47D breast cancer cell lines. Furthermore, biofabricated silver nanoparticle containing water extract of mangosteen bark was reported to preferentially killed A549 lung cancer cells (Zhang & Xiao, 2018). In addition, Kim, Chin & Lee (2017) showed that the mixture of α- and γ-mangostin can inhibit pancreatic cancer cell lines. Their action were contributed by possible autophagy development via AMP-activated protein kinase/mTOR and p38 pathways (Kim, Chin & Lee, 2017). Interestingly, both compounds, together with a common drug called gemcitabine, were also found to synergistically inhibit the cancer cells (Kim, Chin & Lee, 2017), highlighting possible drug concoction for better treatment efficacy.

Anti-microbes

Extracted total xanthones from mangosteen has been shown to possess considerable anti-bacterial and anti-fungal activities (Table 1). Lalitha et al. (2017) showed that ethyl acetate extract of mangosteen leaf was able to inhibit the growth of various bacteria (Staphylococcus epidermidis, Staphylococcus aureus, Micrococcus luteus, Enterobacter aerogenes, Escherichia coli, Vibrio parahaemolyticus, Proteus vulgaris, Klebsiella pneumoniae, Yersinia enterocolitica, and Salmonella typhimurium) and fungi (Trichophyton mentagrophytes 66/01 and T. rubrum 57/01). Nanosized mangosteen pericarp extract has also been shown to possess anti-bacterial properties against Staphylococcus aureus, Bacillus cereus and Shigella flexneri (Sugita et al., 2017). Furthermore, both water (1 mg/mL) and methanol (8.75 μg/mL) extracts of mangosteen pericarp were able to reduce interleukin-6 (IL-6) cytokine production in whole human blood assay infected with Escherichia coli (Elisia et al., 2018), suggesting that the extracts may not just kill bacteria but also act as an anti-inflammatory agent in humans. As such, mangosteen extract has also been used in products related to wound healing. For example, Panawes et al. (2017) demonstrated that gauze coated with both sodium alginate and mangosteen extract was able to inhibit gram positive bacteria including Staphylococcus aureus ATCC 25923 and ATCC 43300 as well as Staphylococcus epidermidis ATCC 12228.

Singly isolated compounds from mangosteen have also been implicated in anti-bacterial activity. For instance, Phuong et al. (2017) showed that α-mangostin acts as a bactericide to Staphylococcus aureus strains including one methicillin resistant Staphylococcus aureus strain which is known to be highly virulent and anti-biotic resistant. Moreover, the compound (α-mangostin) was able to inhibit the bacterial biofilm generation, in particular during its early stage formation. Similarly, various Staphylococcus spp. isolated from bovine mastitis were found susceptible to α-mangostin (minimum inhibitory concentration (MIC) = 1–32 μg/mL) treatment (Chusri et al., 2017), suggesting wide inhibitory action of the compound toward staphylococci strains.

Interestingly, α-mangostin also has been conjugated or modified to be more soluble and potent against bacteria/fungi. Phunpee et al. (2018) revealed that α-mangostin forming inclusion complex with quaternized β-CD grafted-chitosan was able to inhibit Streptococcus mutans ATCC 25177 and Candida albicans ATCC 10231 growth with MIC values of 6.4 and 25.6 mg/mL, respectively. The soluble inclusion complex also possessed higher anti-inflammatory response than free α-mangostin (Phunpee et al., 2018), suggesting solubility may be critical in determining the compound effectiveness. Furthermore, α-mangostin has also been synthetically modified to several analogs particularly at the functional phenolic and iso-prenyl hydroxy groups (Narasimhan et al., 2017). These analogs possessed higher anti-bacteria (against Escherichia coli, Staphylococcus aureus, Bacillus subtilis and Pseudomonas aeruginosa) and anti-fungi (against Candida albicans and Aspergillus niger) activities compared to the original α-mangostin. This highlights the potential use of mangosteen derived compounds in various human applications to curb pathogen infection.

For instance, mangosteen extracts have been commonly used to protect and promote dental health by eradicating oral pathogens. Pribadi, Yonas & Saraswati (2017) showed that the ethanol extract of mangosteen pericarp was able to inhibit the activity of the glucosyltransferase enzyme from Streptococcus mutans, which is important for dental caries progression. Chloroform extract of the same tissue was also shown to be effective against the growth of Streptococcus oralis, Streptococcus salivarius, Streptococcus sanguis and Streptococcus mutans, which are the common pathogens causing dental caries (Janardhanan et al., 2017). Combination of α-mangostin (five mg/mL) and lawsone methyl ether (2-methoxy-1,4-naphthoquinone) (250 μg/mL) has been shown to be effective against oral pathogens such as Streptococcus mutans, Candida albicans, and Porphyromonas gingivalis (Nittayananta et al., 2018). Furthermore, mangosteen extract including α-mangostin has been used as an anti-bacterial component in an adhesive paste to prevent dental caries (Sodata et al., 2017) as well as in a topical gel to cure chronic periodontitis (Hendiani et al., 2017; Mahendra et al., 2017). Interestingly, mangosteen not only kills oral pathogens but also mediate anti-inflammatory response in dental complications. For instance, mangosteen extract has also been shown to reduce inflammation related to gingivitis in rats (Putri, Darsono & Mandalas, 2017). Kresnoadi et al. (2017) further showed that the total extract of mangosteen pericarp could reduce the inflammation of post-tooth extraction in guinea pigs (Cavia cobaya). This can be attributed by the extract ability to lower the protein expression of nuclear factor κβ (NfkB) and receptor activator of nuclear factor-κβ ligand in the treated group (Kresnoadi et al., 2017). These lines of evidence emphasize the use of mangosteen extract in promoting oral hygiene.

Another human application of mangosteen extract is for promoting gastrointestinal health. The growth of probiotic bacteria such as Lactobacillus acidophilus has been shown to be promoted by methanol extract of mangosteen pericarp (Nanasombat et al., 2018). Interestingly, the chloroform extract inhibited the bacteria growth (Janardhanan et al., 2017), suggesting the differences in compounds extracted between more polar (methanol) and lesser polar (chloroform) solvents. However, these studies did not further elucidate the exact compounds from their extracts.

Additionally, compounds from mangosteen may not only restrict bacterial and fungal growth, but also viral infection. For example, α-mangostin has been shown to inhibit dengue virus including all four serotypes (DENV1-4) in infected HepG2 cell lines (Tarasuk et al., 2017). Furthermore, the expression of several chemokine (Regulated upon Activation Normal T cell Expressed and Secreted (RANTES), Macrophage Inflammatory Protein-1β (MIP-1β) and Interferon-inducible protein 10 (IP-10)) and cytokine (IL-6 and tumor necrosis factor (TNF-α)) genes were significantly suppressed in those infected cell lines when treated with α-mangostin (Tarasuk et al., 2017), suggesting that the compound may also mediate inflammatory response upon infection. Meanwhile, malarial parasites, Plasmodium falciparum 3D7 was also inhibited by hexane and ethyl acetate fractions of mangosteen (Tjahjani, 2017), further strengthening the anti-pathogenic use of this plant.

Anti-diabetes

Mangosteen plant extract is known to possess anti-diabetic properties. A nationwide survey in Philippines suggests that the use of mangosteen as tea (pericarp) or eaten raw (aril) could potentially curb diabetes amongst the local population (Mina & Mina, 2017). Although a more thorough clinical trials on human should be conducted, a plethora of recent research in vitro (Table 1) and in vivo (Table 2) have shown that mangosteen extract prospective use for anti-diabetic medication.

For example, various xanthones from mangosteen have been examined with inhibitory activity against certain enzymes or biochemical processes related to obesity. For instance, garcinone E demonstrated strong inhibitory activity against fatty acid synthase enzyme, which is highly expressed in both obese human adipocytes and cancerous cells (Liang et al., 2018). Moreover, two newly discovered xanthones from mangosteen called mangostanaxanthones III and IV prevented advanced glycation end-product, a process where proteins are added with sugars commonly occurring in diabetic cases (Abdallah et al., 2017). Total mangosteen extract has also shown promising result by inhibiting the glycation process in vitro (Moe et al., 2018) as well reducing the activity of digestive enzymes such as α-amylase and cholesteryl ester transfer protein (Mishra, Kumar & Anal, 2016). Furthermore, using an in silico approach, several mangosteen compounds such as 1,3,5,6-tetrahydroxyxanthone, euxanthone, and epicatechin were discovered to be lead compounds for inhibiting pancreatic cholesterol esterase, an important enzyme for hypercholesterolemia, a common syndrome associated with diabetes (Varghese et al., 2017). These highlight potentially specific anti-diabetic drugs from mangosteen could be further developed in the future.

Several in vivo studies to measure mangosteen effectiveness in ameliorating diabetes have also been conducted (Table 2). For instance, diabetic mice supplied with mangosteen vinegar rind (MVR) containing 69% α-mangostin for 1 week were showing relatively lower plasma glucose, total cholesterol, and low density lipoprotein (LDL) levels compared to non-treated diabetic control (Karim, Jeenduang & Tangpong, 2018). Similarly, As’ari & Asnani (2017) showed that mangosteen pericarp extract was able to reduce LDL level in hypercholesterolemia male rats. Furthermore, MVR treatment reduced the levels of hepatotoxic enzymes in the diabetic mice, aspartate aminotransferase and alanine aminotransferase protecting liver from further damage (Karim, Jeenduang & Tangpong, 2018). Moreover, xanthone extract containing 84% α-mangostin prevented triglyceride accumulation in the liver of high fat diet rats, thus avoiding hepatosteatosis complications related to diabetes (Tsai et al., 2016). This hepatoprotective benefit may be resulted from the anti-oxidant capacity of such xanthone extract, as seen by the lower level of reactive oxygen species (ROS) in the treated primary hepatocyte, possibly via the activation of anti-oxidant enzymes including glutathione, glutathione peroxidase, glutathione reductase, superoxide dismutase (SOD), and catalase (Tsai et al., 2016). Furthermore, diabetic mice treated with pure xanthone also improved kidney function by reducing malondialdehyde level, an oxidative stress indicator to prevent kidney hypertrophy (enlargement) (Karim, Jeenduang & Tangpong, 2016). These findings advocate that the mangosteen extracts are not only useful in treating hyperglycemia, but also promoting both liver and kidney health in diabetic patients by way of ameliorating cellular oxidative stress.

Various organ protection

Mangosteen fruit extract has been shown to possess high anti-oxidant level (Chatatikun & Chiabchalard, 2017) as well as anti-inflammatory potential (Fu et al., 2018), which can protect organs such as liver, skin, joint, eye, neuron, bowel, and cardiovascular tissues from damages and disorders.

Mangosteen compounds have been demonstrated to protect liver damage from drug toxification and oxidative stress. For example, acetaminophen (APAP) drug is known to metabolized to a harmful substance that can increase oxidative stress of patients if taken excessively (Fu et al., 2018; Yan et al., 2018). However, xanthones have been shown to attenuate the toxicity and damage on the liver cells of mice by preventing NfkB and MAPK activation, thereby reducing the inflammation on the liver (Ibrahim et al., 2018c; Yan et al., 2018). For instance, α-mangostin prevented the increase of pro-inflammatory cytokines such as IL-6, Interleukin-1β, and TNF-α after treatment with APAP and inhibited the increase of nitric oxide synthase (iNOS) expression, which further protects the liver tissue (Fu et al., 2018). Furthermore, isogarcinol has been shown to possess anti-oxidant activity, without cytotoxic and genotoxic effects on HepG2 liver cells (Liu et al., 2018). The compound also protects those cells from oxidative damage by H2O2, perhaps by increasing anti-oxidant enzymes such as SOD and glutathione as well as reducing the level of active Caspase-3 important for apoptosis (Liu et al., 2018). Wang et al. (2018a) further showed that another major compound from mangosteen, γ-mangostin also exhibited hepatoprotective ability. The compound induced the expression of nuclear factor erythroid 2-related factor 2 (NRF2) which is known to regulate many anti-oxidative enzymes such as heme oxygenase-1 and SOD2. Additionally, γ-mangostin also increased the expression of silent mating type information regulation 2 homologs 1 (SIRT1) which is important for maintaining cellular oxidative stress, in particular reducing ROS production from mitochondrial activity. The action of γ-mangostin in regulating both NRF2 and SIRT1 has been shown in both human hepatocyte cell line L02 induced by oxidants (tert-butyl hydroperoxide) as well as in mice treated with carbon tetrachloride toxic drug (Wang et al., 2018a), suggesting the applicability of this compound in ameliorating liver toxification and oxidative damage.

α-mangostin has also been shown to prevent skin damage and wrinkling due to ultraviolet B (UVB) radiation in hairless mice (Im et al., 2017). The compound acts by reducing matrix metalloproteinases (MMPs) expression, which are the collagen degradation enzymes as well as ameliorating ROS production and inflammation in UVB damaged skin (Im et al., 2017). Furthermore, β-mangostin from mangosteen was able to reduce tyrosine and tyrosinase-related proteins 1 levels to induce depigmentation for skin whitening (Lee et al., 2017). Another mangosteen compound, isogarcinol was shown to be effective against psoriasis (skin lesion) in mice, possibly through mediating pro-inflammatory factors and cytokines (Chen et al., 2017). These suggest that compounds from mangosteen may target certain enzymes from melanogenesis, an important regulatory process for skin protection and complexion.

Mangosteen extract particularly α-mangostin has also been primarily investigated as an anti-arthritic substance. Arthritis is a chronic joint disorder mainly caused by inflammation. Pan et al. (2017a) showed that osteoarthritic rats treated with α-mangostin delayed their cartilage loss. This can be attributed to the compound ability to ameliorate apoptosis and inflammation responses in the cartilage chondrocyte cells as observed by the inhibition of NfkB expression and other IL-1β induced proteolytic enzymes such as MMP-13 and A Disintegrin And Metalloproteinase with Thrombospondin type 1 motifs, member 5 (ADAMTs-5) (Pan et al., 2017a, 2017b). Both Collagen II and Aggrecan proteins were also preserved in α-mangostin treated chondrocytes-induced degradation (Pan et al., 2017a, 2017b). In rheumatoid arthritis, α-mangostin could reduce fibroblast-like synoviocytes which play significant role in joint deprivation (Zuo et al., 2018). This was again due to the action of α-mangostin against NfkB which reduced the inflammatory signals in the arthritic rats (Zuo et al., 2018).

Recently, mangosteen extract has also been shown to improve macular diseases. Treatment with α-mangostin was able to increase SOD and glutathione peroxidase activities to protect mice retina from oxidative damage as well as preserving the retinal photoreceptor against light damage through inhibition of caspase-3 activity (Fang et al., 2016). Interestingly, α-mangostin also was able to accumulate in the retina, suggesting that the compound could pass the blood-retinal barrier (Fang et al., 2016). This again signifies the applicability of mangosteen extract to be used effectively for human application.

Furthermore, γ-mangostin has also been shown to have some potential against neuronal diseases such as Parkinson. The pretreatment of γ-mangostin onto SH-SY5Y cells was able to reduce apoptotic signals such as p38 MAPK phosphorylation and caspase-3 activity from an inducer of Parkinson, 6-hydroxydopamine (Jaisin et al., 2018). The pretreatment also well-preserved the cell viability by reducing the oxidative damage (Jaisin et al., 2018). Similarly, mangosteen extract containing both a- and γ-mangostin can be potentially used for anti-depressant due to its anti-oxidant ability as depression often leads to redox imbalance. The treatment of 50 mg/kg mangosteen extract onto the model animal of depression, flinders sensitive line rats was able to improve cognitive ability and promote the repair process of hippocampal damage of the rats (Oberholzer et al., 2018). α-mangostin also has been modified such that it can inhibit acid sphingomyelinase effectively which is often associated with central nervous system damage and metabolic disorder (Yang et al., 2018). This modified α-mangostin contains C10 hydrophobic tail extension which confer the potency of the compound, is also implicated in anti-inflammatory and anti-apoptotic action against an in vitro NIH3T3 fibroblast cell line treatments (Yang et al., 2018).

Bowel disease including ulcerative colitis is also shown treatable by applying mangosteen extract. For example, ethanol extract of the fruit pericarp containing 25% α-mangostin was able to lower the level of inflammatory proteins such as NfkB of which resulted in the reduction of colitis disease score in mice (Chae et al., 2017). Another report further suggested that the α-mangostin was widely distributed and retained longer in the colon of the treated mice, further increasing its efficacy in the colitis treatment (You et al., 2017).

Another study showed that hypertensive rats with high blood pressure and cardiovascular problems induced by a chemical called Nω-Nitro-l-arginine methyl ester was attenuated by mangosteen extract (200 mg/kg) daily treatment (Boonprom et al., 2017). Such a treatment also reduced the expression of NADPH oxidase subunit p47phox expression responsible for ROS generation, iNOS as well as other pro-inflammatory cytokines such as TNF-α (Boonprom et al., 2017). An in vitro study using hypoxic-induced H9C2 rat cardiomyoblast cells further confirms xanthone roles, in particular α-mangostin to ameliorate oxidative and apoptotic events in cardiac injury (Fang, Luo & Luo, 2018). Furthermore, procyanidin extracted from mangosteen was able to rescue H2O2-treated human umbilical vein endothelial cells (Qin et al., 2017) while xanthones could protect red blood cells from severe H2O2 stress (Buravlev et al., 2018). This suggests that mangosteen extracts may not only protect the structural endothelial cells but also the components of blood vessels (red blood cells).

Interestingly, while mangosteen may contain a plethora of medicinal benefits, one recent study showed that it may act as anti-fertility substance. α-mangostin loaded into nanostructured lipid carriers has been shown to induce spermatogenic cell death and apoptotic Caspases 3/7 activities in testicular tissues of castrated cats (Yostawonkul et al., 2017). Even so, the complex also prevented cellular inflammation through reduced nitric oxide and TNF-α production; such a strategy can be used as a chemical-based animal contraception (Yostawonkul et al., 2017).

Conclusion

This review has covered recent articles related to mangosteen research particularly its compound profile as well as medicinal benefits. Evidently, many mangosteen bioactivities and medicinal benefits are contributed by the presence of phenolic compounds such as xanthones and procyanidins. These compounds are particularly effective against oxidative damage and inflammatory response. As such, mangosteen compounds were able to inhibit cancer and bacterial growth as well as protecting various organs such as liver, skin, joint, eye, neuron, bowel, and cardiovascular tissues from disorders. Despite these benefits, mangosteen compounds have yet to be developed as prescription drugs and hence future effort in human applications should be emphasized.

Supplemental Information

Supplemental Information 1 The number of publications related to mangosteen has increased dramatically in recent years.

Statistics were obtained from SCOPUS database on July 2018 by searching “mangosteen AND Garcinia mangostana” in the “Article title, Abstract and Keywords” search field. Please refer to Supplementary File 1 for the raw data.

Click here for additional data file.

Supplemental Information 2 Countries around the world have published manuscripts related to mangosteen.

Statistics were obtained from SCOPUS database on July 2018 by searching “mangosteen AND Garcinia mangostana” in the “Article title, Abstract and Keywords” search field.

Click here for additional data file.

Supplemental Information 3 Import and export values from various countries around the world for mango, mangosteen and guava.

Data obtained from the Food and Agriculture Organization of the United Nations (FAO) accessed on November 2018 (www.fao.org).

Click here for additional data file.

Supplemental Information 4 Raw data of the statistics obtained using Scopus for the number of publications and countries related to mangosteen research.

Statistics were obtained from SCOPUS database on July 2018 by searching “mangosteen AND Garcinia mangostana” in the “Article title, Abstract and Keywords” search field.

Click here for additional data file.

Additional Information and Declarations

Competing Interests

Author Contributions

Data Availability

The authors declare that they have no competing interests.

Wan Mohd Aizat conceived and designed the experiments, performed the experiments, analyzed the data, contributed reagents/materials/analysis tools, prepared figures and/or tables, authored or reviewed drafts of the paper, approved the final draft.

Ili Nadhirah Jamil analyzed the data, prepared figures and/or tables, authored or reviewed drafts of the paper, approved the final draft.

Faridda Hannim Ahmad-Hashim prepared figures and/or tables, authored or reviewed drafts of the paper, approved the final draft.

Normah Mohd Noor analyzed the data, contributed reagents/materials/analysis tools, authored or reviewed drafts of the paper, approved the final draft.

The following information was supplied regarding data availability:

The raw data is available in the Supplementary File.

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
