# Peer review of "Recent updates on metabolite composition and medicinal benefits of mangosteen plant"

_PeerJ, doi:10.7717/peerj.6324_

## Round 0.1 · original submission · Minor Revisions

Dear authors,

Minor revisions are required for this manuscript. Please, answer point by point, all the comments of reviewers.

Reviewer 1 ·

Basic reporting

the author described carefully review of this caption in order to easily understand the readers.

Experimental design

if possible use any indexing abstract except google scholar and Scopus such as EBSCO, pubmed etc for collecting references of mangosteen related to health and functional compound

Validity of the findings

no comment

Additional comments

The authors should be adding more references about how to increase the functional compounds by agronomical/breeding/etc technicals in their part of the review.

Annotated reviews are not available for download in order to protect the identity of reviewers who chose to remain anonymous.

Reviewer 2 ·

Basic reporting

Well written in English.
Literature nicely referenced.
Figs. can be improved-as commented below.

Experimental design

Article content-nicely investigated with complete details.

Validity of the findings

Well summarized results in a nice table.

Additional comments

Comments on manuscript # 30985 " Recent updates on metabolite composition and medicinal benefits of mangosteen plant" by Aizat et. al.

The author compiled the updated literature about the Mangosteen plant’s medicinal and other research based benefits in a review-nice compilation of all the data at one place.

Following things need to be addressed in the MS.
1. Best thing about this review-The literature was nicely summarized with respect to all the extracts or compounds extracted from the Mangosteen plant up to date. But, the Figure 2-publication record over the years is making no scientific knowledge improvement. I would suggest, author should make a figure showing the different forms of extracts obtained from the Mangoosteen (plant or a fruit) and then all the well characterized compounds from each extract, if any.
2. Related to comment 1, also author can give another figure- very well characterized compounds, e.g., alpha-, beta- , gamma-mangostin, and Xanthones- their structures, if known?

3. Table 1: Author summarized the studies as different categories like anti-cancer, anti-diabetic… excellent job. To make it more effective for all the researchers working in the related area- separate table for in-vivo studies could be even more effective in highlighting the importance of the extracts/compounds for a particular purpose with known dose of the compound/extract used.

The paper is technically sound and well written in nice English. Comments above- will make the MS better.

·

Basic reporting

.

Experimental design

.

Validity of the findings

.

Additional comments

17- October-2018
Journal: Peer J

Manuscript ID:
Title: Recent updates on metabolite composition and medicinal benefits of mangosteen plant

Authors: Wan Mohd Aizat, Faridda Hannim Ahmad-Hashim, Normah Mohd Noor

Dear Editor:

The authors have investigated the chemical compositions and medicinal benefits of mangosteen plant. The authors present a sufficient interesting review and thus I accept this review for publication after minor revision.
.
Hesham El-Seedi, Professor,
Division of Pharmacognosy
Department of Medicinal Chemistry
Uppsala University, Biomedical Centre
Box 574, SE-75 123, Uppsala, Sweden
Tel: +46-18-4714207
Fax: +46-18-509101
E-mail: [email protected]


Comments to Author:
List of authors:
1-the authors would add the affiliations of co-authors.
Abstract:
1- The authors would add some examples of the uses of the plant in folk medicines
2- The authors could show the known medicinal benefits.

Introduction:
1- The authors would add references for line 41:45, 48:50 and 51:54 in page 7.
2- The authors would clarify effect of this plant that used in folk medicines compared to other plants or solely.
3- The authors would clarify this sentence (Such claim is not without experimental basis as shown in various studies to characterise the fruit usages as anti-oxidant, anti-inflammatory, anti-cancer and anti-hyperglycemic substance).
Metabolite composition of mangosteen:
1- The authors would check again the statements without references.
2- The authors would show pharmaceutical properties of the compounds.
3-If the authors would need more information about xanthones, take benefit from the below mentioned reviews:
1- El-Seedi, H.R., El-Barbary, M.A., El-Ghorab, D.M.H., Bohlin, L., Borg Karlson, A.-K., Göransson, U., Verpoorte, R. (2010). Recent insights into the biosynthesis and biological activities of natural xanthones. Current Medicinal Chemistry 17, 854-901.
2- El-Seedi, H.R., El-Ghorab, D.M.H., El-Barbary, M.A., Zayed, M.F., Göransson, U., Larsson, S., Verpoorte, R. (2009). Naturally occurring xanthones; latest investigations: isolation, structure elucidation and chemosystematic significance. Current Medicinal Chemistry 16, 2581-2626.
4- The authors could put the ratio of chemical compounds in figure and general chemical structures.
5-the authors could use (anti-microbial) instead of (Anti-bacteria, anti-fungi and anti-virus)
Conclusions:
Conclusion should follow the instructions of the journal.
References:
The authors should follow the instructions of the journal and add DOI and link of journal.

---

## Round 0.2 · accepted · Accept

Dear Wan,

Thank you for your submission to PeerJ.

I am writing to inform you that your manuscript - Recent updates on metabolite composition and medicinal benefits of mangosteen plant - has been Accepted for publication.

Congratulations!

Reviewer 1 ·

Basic reporting

no comment

Experimental design

no comment

Validity of the findings

no comment

Additional comments

no comment

Reviewer 2 ·

Basic reporting

Standard English, well cited literature and nice figures/tables reported.

Experimental design

Well defined methods and content.

Validity of the findings

Nice conclusions and well written summary.

Additional comments

The author included the comments of all reviewers nicely in their revised MS and thus the MS looks better now.

·

Basic reporting

Yes, it has been modified according to our suggestions.
I strongly recommend the paper for publication.

Kindest regards, Hesham

Experimental design

Yes, it has been modified according to our suggestions.
I strongly recommend the paper for publication.

Kindest regards, Hesham

Validity of the findings

Yes, it has been modified according to our suggestions.
I strongly recommend the paper for publication.

Kindest regards, Hesham

Additional comments

Yes, it has been modified according to our suggestions.
I strongly recommend the paper for publication.

Kindest regards, Hesham